# The History of Nerve Growth Factor: From Molecule to Drug

**DOI:** 10.3390/biom14060635

**Published:** 2024-05-29

**Authors:** Elizabeth Gavioli, Flavio Mantelli, Maria Candida Cesta, Marta Sacchetti, Marcello Allegretti

**Affiliations:** 1Dompé U.S. Inc., 181 2nd Avenue, STE 600, San Mateo, CA 94401, USA; elizabeth.gavioli@dompe.com; 2Dompé farmaceutici S.p.A., Via Santa Lucia, 6, 20122 Milano, Italy; flavio.mantelli@dompe.com (F.M.); candida.cesta@dompe.com (M.C.C.); marta.sacchetti@dompe.com (M.S.)

**Keywords:** biomolecule, cenegermin, drug discovery, nerve growth factor, neurotrophins, neurotrophic keratitis

## Abstract

Nerve growth factor (NGF), the first neurotrophin to be discovered, has a long and eventful research journey with a series of turning points, setbacks, and achievements. Since the groundbreaking investigations led by Nobel Prize winner Rita Levi-Montalcini, advancements in the comprehension of NGF’s functions have revolutionized the field of neuroscience, offering new insights and opportunities for therapeutic innovation. However, the clinical application of NGF has historically been hindered by challenges in determining appropriate dosing, administration strategies, and complications related to the production process. Recent advances in the production and scientific knowledge of recombinant NGF have enabled its clinical development, and in 2018, the United States Food and Drug Administration approved cenegermin-bkbj, a recombinant human NGF, for the treatment of all stages of neurotrophic keratitis. This review traces the evolutionary path that transformed NGF from a biological molecule into a novel therapy with potential research applications beyond the eye. Special emphasis is put on the studies that advanced NGF from discovery to the first medicinal product approved to treat a human disease.

## 1. Introduction

“The entire history of nerve growth factor (NGF) can be compared to the discovery of a sunken continent revealed by its emerging top [1].” With these words taken from the book *Elogio dell’imperfezione*, Rita Levi-Montalcini describes her discovery and contribution to medical science [1]. In this review, we discuss how researchers are discovering that supporting neuronal growth is just the tip of the iceberg regarding the physiological role of NGF.

NGF was the first discovered member of the neurotrophin family [2]. Other members include brain-derived neurotrophic factor (BDNF), neurotrophin 3 (NT-3), and neurotrophin 4 (NT-4) [3]. The discovery of NGF traces back to the 1950s, thanks to the collaborative efforts of Nobel Prize winners Rita Levi-Montalcini and Stanley Cohen at Washington University [4]. The discovery of NGF marked a significant milestone, and Dompé farmaceutici S.p.A. played an integral role in the development of NGF into a medicinal product.

NGF is an insulin-like protein that exerts its biological action via two receptors: tropomyosin receptor kinase A (TrkA), with tyrosine kinase activity, and P75 neurotrophic receptor (p75NTR), a transmembrane glycoprotein, also called “high-affinity receptor” and “low-affinity receptor”, respectively [5]. While all neurotrophins bind the p75NTR receptor, the interaction with the Trk receptor is specific; NGF binds TrkA, BDNF binds TrkB [6], and NT-3 binds TrkC [7,8]. Once the ligand is bound to this family of receptors, activation of the tyrosine kinase domain occurs, resulting in an increase in the phosphorylation of substrates that leads to different biological functions [9], such as the development and maintenance of neurons, regulation of wound healing, and differentiation and survival of immune cells [10]. The effects of NGF depend on the co-distribution of these receptors on the cell surface [11].

The synthesis of NGF can occur in many cell types [10]. NGF is synthesized as a precursor, proNGF, which can be cleaved intracellularly into mature NGF by furin [12], extracellularly by plasmin or matrix metalloproteinases [13], or remain and signal in its precursor form [14]. The crystal structure of NGF was determined in 1991 using X-ray crystallography [15], revealing important details about its three-dimensional arrangement. NGF comprises two monomers interconnected noncovalently through a cysteine knot structure, resulting in a twisting of beta chains around one another.

As mentioned previously, NGF is synthesized and has different biological functions in various cell types [10]. Scientific contributions from Levi-Montalcini include the effect of NGF on cell types other than sympathetic and sensory neurons, the influence of NGF on both the central nervous system (CNS) and the immune system, the ability of NGF to act as a chemotropic agent, and overexpression of NGF in the bloodstream and hypothalamus during stressful events [4]. Since NGF is also active beyond the nervous system, it is recognized as a pleiotropic factor [16,17]. NGF is produced and used by several cell types including structural cells (epithelial and endothelial cells, fibroblasts, smooth muscle cells, hepatocytes), accessory cells (Müller cells, astrocytes), and immune cells (eosinophils, granulocytes, lymphocytes, mast cells, antigen-presenting cells) [18]. The various functional activities of NGF on these cell types include activation, differentiation, proliferation, survival, and migration that occur via crosstalk with other soluble factors including cytokines, neuropeptides, and other growth factors. The discovery of NGF is regarded by neurobiologists as the first identification of a class of trophic molecules that serve as a regulatory link between a target and innervating nerve cells [4]. An overview of NGF-related upstream and downstream pathways is presented in Figure 1.

With its multiple properties, NGF offers significant promise for treating a wide range of human diseases. The ability to impact the growth, survival, and activation of neurons makes it a promising candidate for therapeutic applications [18]. For many years, mouse NGF was regarded as the gold standard for studying bioactivity. Its close evolutionary relationship to human NGF, ready availability, and assumption of identical properties to its human counterpart led to extensive testing and evaluation in laboratory practice and preclinical models, as well as scrutiny in academic clinical studies [20].

### 1.1. Background: First Studies and Discovery of NGF

The history of NGF goes back to 1949, when Rita Levi-Montalcini saw the results from Elmer Bueker, which showed that the implantation of a small fragment of malignant mouse tumor tissue into chick embryos resulted in the invasion of sensory fibers into the mouse tumor [4]. Levi-Montalcini investigated the effects of transplanting two mouse sarcoma tissues into a chick embryo and found that the tumor tissues induced hyperinnervation of internal organs. She then hypothesized that transplanted tissues released a diffusible agent that stimulated the growth and differentiation of developing nerve cells. At the beginning of the 1950s, she and Stanley Cohen conducted several experiments to characterize the properties of this factor (NGF) by using snake venom to determine whether the active molecule was a nucleic acid or a protein. They then discovered a rich source of NGF in the mouse gland [21,22] and successfully isolated and purified the molecule in 1960, identifying it as a protein [23,24].

After NGF was characterized and purified, subsequent studies aimed to understand its properties and functions. These studies led to significant discoveries about the multiple roles of the protein, including its importance in the development of sympathetic and sensory ganglia, effects on other cell types, influence on the CNS, capacity as a chemotropic agent, action on cells of the immune system, and role in homeostatic responses [25]. 

Due to the abundant levels and straightforward purification of biologically active NGF from the submandibular glands of male mice, murine NGF (mNGF) was used in various laboratory experiments and in select preclinical and clinical investigations [26]. When comparing murine and human NGF, significant differences emerge in biochemical and biophysical properties. Notably, both mNGF and mproNGF exhibit lower stability in proteolytic cleavage and chemical and thermal denaturation than their human counterparts. Furthermore, mNGF shows reduced potency in its interaction with the human TrkA receptor as observed in the TF1 cell proliferation assay [27]. Additionally, the low sequence conservation in the propeptide domain of their precursor proteins indicates variation in the biological functions of mNGF and human NGF [27].

### 1.2. Potential Applications of NGF

So far, NGF has attracted significant research attention because of its potential applications in various therapeutic fields: neurodegenerative diseases such as Alzheimer’s disease (AD), Parkinson’s disease (PD), amyotrophic lateral sclerosis [28,29,30,31], spinal cord injury [32], and peripheral nerve injury [33]; Rett syndrome [33]; psychiatric disorders including depression, anxiety, and post-traumatic stress disorder [34,35]; and neurodevelopmental disorders such as autism spectrum disorder and attention deficit hyperactivity disorder [36,37]. Targeting the NGF pathway has been suggested in pain management for chronic conditions such as neuropathic pain [38,39,40]. The use of NGF as a therapeutic agent has also been considered for neuronal survival, stroke rehabilitation [41], and the treatment of ocular disorders such as glaucoma and age-related macular degeneration [42,43].

The potential applications of NGF have expanded beyond the nervous system, making it a “pleiotrophic molecule” to include various tissues and organs such as skin and parenchymal organs [44,45,46]. Targeting the NGF pathway has also been suggested in inflammatory disease, male infertility, and cancer [47,48,49]. However, it is important to note that while its therapeutic potential is vast, the clinical usefulness of NGF has not been fully explored, and nonophthalmic uses are still being validated. As confirmation of this, it is noteworthy to report some preliminary data on the use of NGF in CNS diseases, like PD and AD. A 63-year-old woman with a 19-year history of PD with drug-induced complications (hyperkinesia, on–off phenomenon) experienced sustained functional improvement after receiving an intraputamenal infusion of NGF for 23 days [31]. She experienced no safety concerns and was able to extend the duration of adrenal chromaffin graft effects for at least 11 months. Similarly, the use of an intraventricular infusion of NGF in patients with AD [50,51] suggested a potential improvement in neuropsychological effects. However, side effects such as back pain were described with the intraventricular administration route, necessitating further clinical studies on alternative routes of administration along with different dosages of NGF to mitigate adverse reactions. 

#### 1.2.1. Potential Applications of NGF: Wound Healing

NGF has also demonstrated its ability to accelerate wound healing rates in preclinical models, including both normal and diabetic mice, suggesting its potential as a therapeutic option for both skin and corneal ulcers in humans. This has been supported in multiple studies in which purified murine NGF demonstrated accelerated wound healing in cutaneous lesions [52,53] as well as in patients with chronic vasculitis secondary to rheumatoid arthritis (RA) with cutaneous ulcers [54], which also led to a significant reduction in local pain and inflammation. Similarly, patients with diabetes-associated chronic ulcers and with damage extending beyond the hypodermis and muscle layers experienced accelerated healing within a few weeks of NGF treatment, with notable reductions in size after 8 weeks [55]. More recently, the use of a modified human recombinant NGF derivative accelerated skin repair in diabetic mice with pressure ulcers induced by repeated ischemia–reperfusion cycles without any safety concerns of systemic absorption [56]. The molecular mechanism targets the protein kinase B/mammalian target of rapamycin pathway, leading to endothelial cells stimulating the proliferation of both keratinocytes and fibroblasts. The rationale behind using NGF to enhance wound healing in diabetes is rooted in the condition’s association with diminished nerve endings in the epidermis [57].

#### 1.2.2. Potential Applications of NGF: Optic Gliomas

At the intersection of ophthalmology and neuro-oncology, optic gliomas (OGs) represent low-grade brain tumors characterized by slow progression with progressive visual loss standing out as one of the primary morbidities [58]. Due to NGF’s neuroprotective properties promoting neural recovery and survival, it was investigated in a randomized, double-blind, placebo-controlled phase 2 trial of 18 patients with optic pathway glioma [59]. A 0.5 mg dose of murine NGF led to a statistically significant improvement in objective electrophysiological parameters (PhNR amplitude and VEP) and patients retaining measurable visual field. There were no notable side effects experienced related to NGF, and importantly, no tumor growth was observed by magnetic resonance imaging (MRI) scans at 6 months. This aligns with previous research, which similarly found no safety concerns associated with short-term NGF treatment, and no tumor growth was observed in the two years following completion of the study [60,61]. This phase 2 study demonstrated the potential of NGF treatment in diminishing the necessity for aggressive treatments and mitigating long-term consequences [62] typically associated with tumor progression and visual decline in OG. A confirmatory phase 3 study is required to support these results.

#### 1.2.3. Potential Applications of NGF: Pain

In addition to its neurotrophic actions, NGF is a pain mediator that has sensitizing functions, mediated by TrkA expressed on mast cells, and proinflammatory functions. Elevated levels of NGF RNA and protein [63] have been obtained from pain models and patients with various pain disorders. Mutations (R121W, V232fs, and R221W) identified in the NGF gene have been linked to Hereditary Sensory and Autonomic Neuropathy type V (HSAN V), a rare genetic disorder characterized by insensitivity to pain and alterations to the autonomic nervous system [64,65]. These mutations alter the balance between NGF and proNGF, leading to an accumulation of proNGF and a potential analgesic target of its downstream signaling pathway. Current treatments for acute and chronic pain have limited efficacy with limited long-term safety concerns. Anti-NGF antibodies, tanezumab, fasinumab, and fulranumab, work by neutralizing NGF bioactivity by binding directly to NGF. They have demonstrated promising results for the treatment of osteoarthritis (OA) [66]; however, adverse effects, such as osteonecrosis, neurogenic arthropathy, rapid-progression OA, and effects on the sympathetic system have placed these medications out of favor. Additionally, the NGF inhibitors exhibited only a modest effect size (−0.26, 95% CI −0.46 to −0.05) [67] which was only slightly better than NSAIDs, and therefore not clinically different than conventional OA treatments. There still remains hope to use the information gained from HSAN V to identify new pathways and therapeutic proteins involved in pain transmission given the remaining unmet need for disease-modifying therapeutics. The development of additional investigational medications targeting NGF continues to move forward in preclinical and clinical development.

Future studies that are randomized, controlled clinical trials are required to demonstrate sufficient evidence for NGF’s therapeutic benefit in each of the disease states mentioned in this section.

### 1.3. Recombinant Human NGF

In 1982, the Swiss-based pharmaceutical company Sandoz, which later merged with Ciba-Geigy to form Novartis, emerged as an early pioneer in recognizing the remarkable potential of NGF for treating a wide range of neurological disorders [4]. Swiss neuroscience professor Beat Gähwiler contributed his expertise, collaborating with the Sandoz research team, to initiate a series of experiments using organotypic cultures, a distinctive model system that faithfully replicates in vivo conditions of neuronal development. The discoveries made in the context of organotypic cultures laid the groundwork for a deeper understanding of NGF’s role in neural development and plasticity. This knowledge would later prove invaluable in developing novel therapeutic approaches for neurological disorders, particularly those associated with cholinergic system dysfunction. In these early stages of NGF research, it quickly became evident that the development of the protein as a drug was premature and complex. Crucial information and material for its development and practical application were not available, including the limited availability of NGF as well as an incomplete understanding of its molecular structure and the receptors responsible for its effects [68].

In 1983, the cloning and expression of NGF at Genentech represented a breakthrough [69]. This enabled the production of substantial protein quantities and eliminated the laborious task of purifying small amounts of NGF obtained from mouse salivary glands. In 1992, Genentech started the development of recombinant human NGF (rhNGF) as a treatment for sensory peripheral neuropathies [69], which would go on to complete phase 2 and phase 3 clinical trials before being suspended and later discontinued [70]. 

#### Recombinant Human NGF: Early Clinical Trials

A phase 2, randomized, double-blind, placebo-controlled clinical trial that evaluated rhNGF for the treatment of HIV-associated sensory neuropathy in 270 patients was completed [71]. Because NGF is essential to the development and maintenance of sympathetic and sensory neurons and their outgrowths [72], the study proposed that rhNGF would provide a specific restorative treatment for painful HIV-associated sensory neuropathy [71]. Patients were randomized to receive either rhNGF at one of two doses (0.1 or 0.3 µg/kg) or placebo administered subcutaneously twice weekly for 18 weeks. Analysis of global assessment of pain showed that more participants treated with rhNGF reported improvement in neuropathic pain at week 12 (rhNGF 0.3 µg/kg, 51%; placebo, 23%; *p* < 0.001) and had improved pain sensitivity compared with participants receiving the placebo (*p* = 0.019), assessed by neurological examination. The mean difference in Gracely pain improvement was significant from baseline to week 18 in the rhNGF 0.3 µg/kg group compared with placebo (0.20 versus 0.05, respectively; *p* = 0.039). The only notable side effect reported was pain at the injection site, occurring in 53/180 (29.4%) of patients receiving rhNGF. When evaluated for safety, no significant treatment effects on plasma HIV RNA levels between baseline and week 18 were observed. No further larger randomized controlled studies or phase 3 clinical trials were conducted in the population with HIV-associated sensory neuropathy.

A phase 3 placebo-controlled clinical trial was also conducted by Genentech to assess the efficacy and safety of rhNGF in patients with diabetic neuropathy [73]. No difference in the primary endpoint (NIS-LL score) was observed, which was defined as improvements in nerve function after 12 months of treatment; in addition, no beneficial effect of rhNGF vs. placebo was observed for the majority of the secondary endpoints; thus, the trial failed to demonstrate a significant beneficial effect of rhNGF on diabetic polyneuropathy, unlike the previous phase 2 trial results. The study’s failure [74] was attributed to several factors: (1) despite being powered on prior research results, the population within the phase 3 trial was substantially different from the previous large epidemiological study, the Rochester Diabetic Neuropathy Study; (2) the rhNGF dose was insufficient to determine efficacy; (3) different patient populations due to modifications in the inclusion and exclusion criteria across clinical trials; and (4) changes in the manufacturing process and formulation prior to the phase 3 trial led to a reduction in the final concentration from 2 mg/mL in the phase 2 trial to 0.1 mg/mL in the phase 3 trial. Based on these results, Genentech decided to discontinue further studies with rhNGF.

### 1.4. From Molecule to Medicinal Product: Ophthalmology Clinical Use

In 2011, Dompé and Anabasis Irl, an Italian biopharmaceutical company founded in 1999 by a group of Italian scientists dedicated to the development of innovative therapies for the eye, agreed to cooperate, and in 2012, Dompé ran the first clinical trial of rhNGF in ophthalmology, specifically for the treatment of moderate-to-severe unilateral neurotrophic keratitis (NK). This was possible because Dompé was already able to produce rhNGF in the quantities required (see Section 1.5).

To date, as a result of preclinical and clinical studies, Dompé markets the first approved rhNGF-based eye drops for treating NK [75]. This drug is the first-ever topical biologic approved in the field of ophthalmology and represents the first approved clinical application of NGF. However, the research efforts have not been limited to this application; the company marketing and producing rhNGF continues to pursue the extension of the use of this neurotrophin to other neurosensory system pathologies and exploring the rationale for the use of other neurotrophins, such as recombinant human BDNF (rhBDNF). Due to the expertise gained by the refinement of the NGF manufacturing process, transitioning to the production of additional neurotrophins may in fact be expedited. 

#### From Molecule to Medicinal Product: Ophthalmology Clinical Use—rhNGF Production

The challenge in rhNGF production is obtaining the active protein in high yields [68]. The first step in producing rhNGF involves synthesizing a gene that codes for the human proNGF protein. This gene is subsequently cloned into an expression vector to create a plasmid, resulting in a strain of the *Escherichia coli* microorganism with the necessary information to express the protein. The “pro” segment is responsible for the specific chaperone of rhNGF and is located at the N-terminal residue to enable proper NGF folding.

A series of fermentations in *E. coli* are then conducted, enabling recombinant protein production [68]. The protein accumulates in the cytoplasm of the microorganism as inclusion bodies. These inclusion bodies can be easily recovered through centrifugation and then broken to recover the protein, which, thanks to the “pro” sequence, is refolded to its native and active form. Dompé has equipped itself with cell banks, including a master cell bank and a working cell bank, which serve as continuous repositories of frozen cells stored at −160 °C in vials to be used when needed for production purposes, ensuring that the process always starts from the same cell source. An overview of the production of rhNGF for subsequent fermentations in *E. coli* up to the purification of the recombinant active protein from the inclusion bodies is shown in Figure 2A. The workflow of upstream and downstream activities for the production of the active ingredient cenegermin-bkbj is described in Figure 2B.

### 1.5. Clinical Trials of rhNGF—Ophthalmology

Table 1 shows a nonexhaustive list of ongoing and completed clinical trials on rhNGF in order of clinical trial phase and in various ophthalmic indications, highlighting the interest in this protein and its clinical applications.

### 1.6. Clinical Trials of rhNGF for Neurotrophic Keratitis

Neurotrophic keratitis, a rare condition (ORPHA:137596) with a prevalence of <5 people per 10,000 in the general population [96,97], is a degenerative disease resulting from impaired corneal sensory innervation and loss of corneal sensation that causes progressive damage to the top layer of the cornea, leading to ulceration and, in severe cases, perforation [98]. 

Dompé has studied the safety and efficacy of a rhNGF topical eye solution in patients with NK in two clinical trials. These trials were planned as randomized, placebo-controlled, multicenter, double-masked studies.

#### 1.6.1. NGF0212, Phase I and II Clinical Trials

NGF0212 (REPARO [European study]: NCT01756456) was a phase 1/2 randomized, double-masked, parallel-group clinical trial aiming to assess the efficacy and safety of two dosing regimens (10 or 20 µg/mL, administered six times daily) of rhNGF eye drop solution compared with a vehicle in stage 2 (persistent epithelial defect) and stage 3 (corneal ulcer) NK over 8 weeks [87] with a 48- or 56-week follow-up period [87,99].

The phase 1 study included 18 patients divided into two cohorts of 9 consecutively enrolled patients with stage 2 or 3 NK [87,99]. They were randomized in a 7:2 ratio with one group receiving rhNGF at 10 µg/mL versus a vehicle (cohort A) and the other group receiving rhNGF at 20 µg/mL versus a vehicle (cohort B). During controlled treatment, eye pain and headache were the most frequently reported treatment-related adverse events (trAEs), with each occurring in two patients (28.6%) in the rhNGF 20 µg/mL group. trAEs were reported by only 1 of 18 patients in each group. Notably, no trAEs were reported during the 48-week follow-up period. As for the pharmacokinetic (PK) profiling, only two patients had detectable serum NGF levels at any time point. Of note, a patient in the rhNGF 10 µg/mL group had only one positive NGF measurement during the study. In comparison, the patient in the rhNGF 20 µg/mL group had detectable serum NGF levels at all time points, even before initiating the study treatment. In summary, the PK results suggest individual fluctuations in endogenous NGF levels independent of the study treatment.

The REPARO phase 1 study demonstrated that topical ophthalmic rhNGF, administered at either 10 or 20 µg/mL with six drops daily for 8 weeks, was well tolerated by patients with stage 2 or 3 NK [99].

The phase 2 study assessed the efficacy and safety of rhNGF at concentrations of 10 µg/mL, 20 µg/mL, or a placebo (vehicle) in 156 patients randomized in a 1:1:1 ratio [87,100]. Six drops were administered daily for 8 weeks, after which patients entered a 48- or 56-week follow-up period. The primary efficacy measure was corneal healing defined as less than 0.5 mm fluorescein staining in the lesion area at week 4 for the primary endpoint (week 8 was assessed as the secondary endpoint). 

Corneal healing was achieved by 19.6% of patients in the vehicle group at week 4, compared with 54.9% in the rhNGF 10 µg/mL group (*p* < 0.001) and 58.0% in the rhNGF 20 µg/mL group (*p* < 0.001) [100]. At week 8, corneal healing was observed in 43.1% of vehicle-treated patients, while 74.5% of those receiving rhNGF 10 µg/mL (*p* = 0.001) and 74.0% of those receiving rhNGF 20 µg/mL (*p* = 0.002) achieved healing. Visual acuity outcomes were assessed by measuring changes from baseline to week 8; more patients in the rhNGF 10 µg/mL and rhNGF 20 µg/mL groups achieved 15-letter gains than patients in the vehicle group. During the controlled treatment period, AEs were reported in 25 patients: 6 in the rhNGF 10 µg/mL group, 9 in the rhNGF 20 µg/mL group, and 10 in the vehicle group. Most AEs were mild, transient, and did not require discontinuation of treatments.

Overall, 17 patients (10.9%) experienced serious AEs during controlled treatment; however, none were related to the study treatment.

The REPARO phase 2 study showed that topical rhNGF safely and effectively improves corneal epithelial integrity in cases of moderate-to-severe NK [100].

#### 1.6.2. NGF0214, Phase II Clinical Trial

NGF0214 (NCT02227147; US study) was a phase 2 pivotal trial aiming to assess the effectiveness of 20 µg/mL rhNGF eye drops (cenegermin-bkbj ophthalmic solution) administered six times daily for 8 weeks of masked treatment (follow-up of 24 weeks) compared with a vehicle containing the antioxidant methionine in patients with stage 2 and 3 NK [84,101]. The study enrolled adult patients (≥18 years of age) with NK in one or both eyes who were required to discontinue all previous topical ophthalmic medications and/or contact lenses upon enrollment [101]. The primary efficacy endpoint was healing of the neurotrophic lesion after 8 weeks of masked treatment. In this pivotal trial, 20 µg/mL rhNGF eye drop solution effectively promoted the healing of persistent epithelial defects (rhNGF, 65.2%; vehicle, 16.7%; *p* < 0.001) in patients with NK at week 8, defined as 0 mm staining in the lesion area and no other persistent staining outside of the lesion area. The most common trAEs included eye pain, foreign body sensation, and tingling, which suggested nociceptor sensitization. In summary, rhNGF eye drop solution is a tolerable, noninvasive, pharmacologic treatment for NK, becoming a part of the treatment algorithm for this often challenging-to-manage disease.

After these positive results, Oxervate^®^ (cenegermin-bkbj ophthalmic solution [20 µg/mL rhNGF]) was granted orphan drug status and received breakthrough therapy designation from the US Food and Drug Administration (FDA) in 2018 for treating all stages of NK [102,103].

The comparison of phase 1/2 (Europe) and phase 2 (United States) clinical trials is shown in Table 2.

## 2. Discussion and Future Perspectives

NGF, originally discovered by the Italian neurobiologist Rita Levi-Montalcini and her US colleague Stanley Cohen [4], has completed several clinical trials with many still ongoing for the treatment of various ocular diseases. Significant progress in research has been made possible by the industrial-scale production of rhNGF. This enabled the initiation of a clinical development project for rhNGF in ophthalmology involving patients with moderate-to-severe unilateral NK [103]. Cenegermin-bkbj ophthalmic solution has proven to be well tolerated and effective in inducing corneal healing in patients with NK.

Currently, Dompé is investigating rhNGF in other ophthalmic conditions, including Sjögren’s dry eye disease, which affects approximately 4 million Americans and is considered a particularly challenging condition to treat because of its complex etiology mediated by autoantibody production and lymphocytic infiltration [105,106]. To explore the full potential of rhNGF, the company is also exploring the possibility of extending the applications of rhNGF for the treatment of neuropathic corneal pain and diseases of other sensory organs. Oxervate^®^ is the first-ever topical biologic medication approved in ophthalmology [103], with research in this therapeutic area continuing to expand [75].

As originally hypothesized by the discoverers of NGF, this pleiotropic molecule holds potential research also beyond the eye. In fact, extensive studies are being carried out on the involvement of NGF in the spinal cord [32], peripheral nerve injuries [33], psychiatric disorders [34,35], and pain management of chronic diseases [38,39,40]; its therapeutic potential in AD [51], PD [31], and wound healing in patients with diabetes [55]; and its role in oncological diseases such as OG [59]. The development of pharmaceutical compounds designed to target NGF, or its receptors, could also be valuable through temporal dosing strategies for the treatment of autoimmune diseases such as osteoarthritis and rheumatoid arthritis in which NGF is significantly upregulated in the phasic progression of inflammation [107] and plays a role in mediating neuropeptides and neurotransmitters to affect innate and adaptive immune responses [108]. There remains an unmet need to investigate compounds that may target mutations in the receptors of Trk and p75NTR, which affect the production of NGF and play a role in autoimmune diseases and inflammation [109]. Lastly, nasal NGF administration has recently shown potential for improving perfusion, metabolism, and brain function in children with traumatic brain injury [110,111,112], further increasing the interest in new therapeutic indications for rhNGF. This method of administration may provide future guidance for other CNS disease states in which self-administration can be used with the benefit of minimal risks of side effects related to CNS delivery.

## 3. Conclusions

Since the discovery of NGF, numerous in vivo and in vitro studies have documented the neuroprotective role of NGF for damaged cells of the nervous system. However, nowadays, there is increasing evidence that supports the potential therapeutic effects of NGF in ocular diseases (corneal ulcer, glaucoma, retinitis pigmentosa, macular degeneration, and diabetic retinopathy), epithelial healing, and certain disorders of the peripheral and CNS. The discovery of NGF marked a significant milestone in the field of neuroscience research, whereas the transformation of NGF into a revolutionary medicinal product with vast potential was enabled by the efforts undertaken by the pharmaceutical industry that stepped in to overcome the challenges of rhNGF production. The successful development of rhNGF from a molecule to a medicinal product represents a great example of what can be achieved by the collaboration between academia and industry.

## Figures and Tables

**Figure 1 biomolecules-14-00635-f001:**
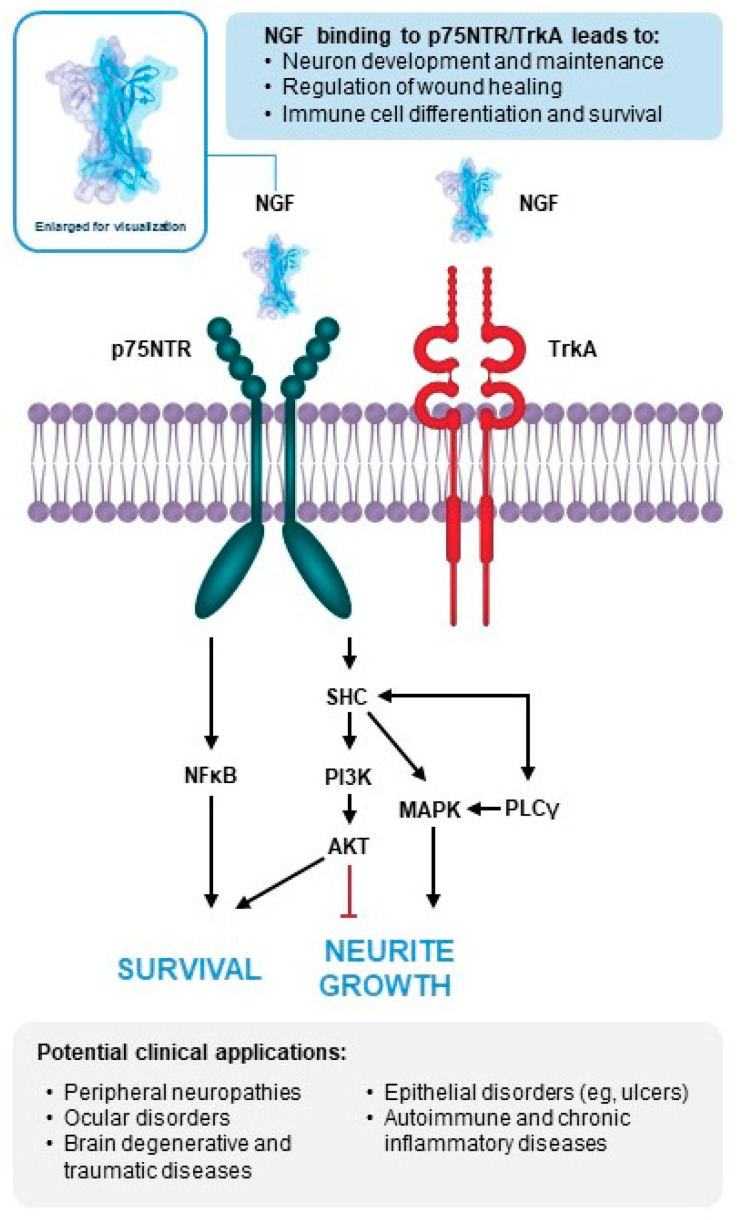
NGF-related upstream and downstream pathways and clinical application overview [10,19]. NGF (shown as a crystal structure comprising 2 monomers interconnected noncovalently through a cysteine knot structure) binds and activates TrkA and p75NTR, resulting in downstream survival pathway activation via NFκB and AKT, inhibition of neurite growth via AKT, and promotion of neurite growth via MAPK. AKT, protein kinase B, c-Jun N-terminal kinase; MAPK, mitogen-activated protein kinase; NFκB, nuclear factor kappa B; NGF, nerve growth factor; p75NTR, p75 neurotrophin receptor; PI3K, phosphoinositide 3-kinase; PLCγ, phospholipase C gamma; SHC, SHC-transforming protein 1; TrkA, tropomyosin receptor kinase A.

**Figure 2 biomolecules-14-00635-f002:**
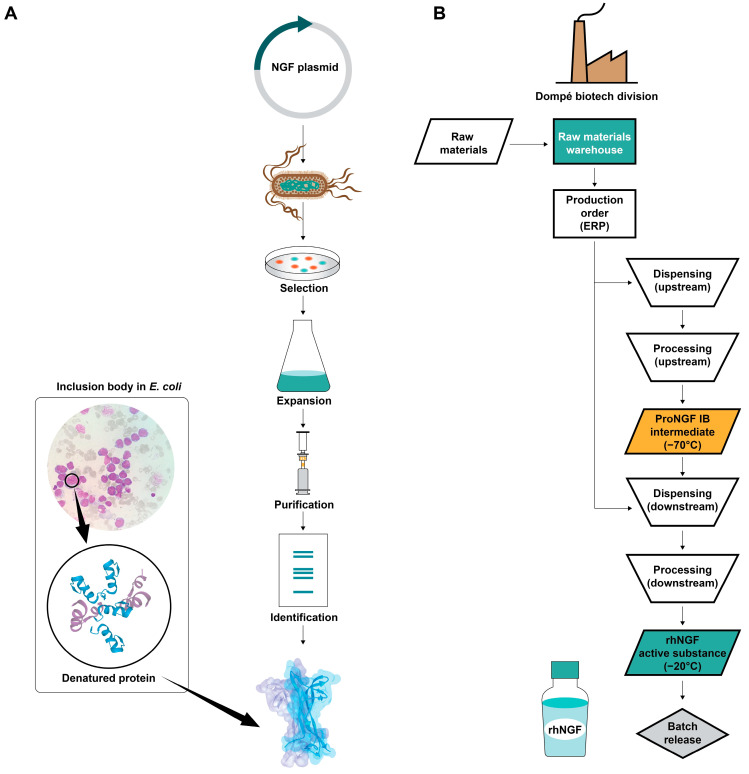
Recombinant human NGF production and cenegermin-bkbj workflow. (**A**) rhNGF production begins with a gene that codes for the human proNGF protein cloned into an expression vector to create a plasmid, resulting in a strain of the *Escherichia coli* microorganism with the necessary information to express the protein. A series of fermentations in *E. coli* are conducted, enabling recombinant protein production. The protein accumulates in the cytoplasm of the microorganism as inclusion bodies, which are recovered through purification. (**B**) Workflow of cenegermin-bkbj production. ERP, enterprise resource planning; IB, inclusion body; NGF, nerve growth factor; rhNGF, recombinant human NGF.

**Table 1 biomolecules-14-00635-t001:** A nonexhaustive list of ongoing and completed clinical trials on the investigational use of rhNGF in ophthalmic diseases.

Identifier	Condition	Treatment	Sponsor	Clinical Trial Phase	Recruitment Status	Start/End Date
NCT05552261 DEFENDO—NGF0122 [76]	Stage 1 neurotrophic keratitis	Cenegermin-bkbj	Dompé farmaceutici S.p.A.	4	No longer recruiting	20 December 2022 4 September 2024
NCT04485546 DEFENDO—NGF0120 [77]	Stage 1 neurotrophic keratitis	Cenegermin-bkbj	Dompé farmaceutici S.p.A.	4	Completed	9 September 2020 30 March 2022
NCT05136170 [78]	Sjögren’s dry eye	Cenegermin-bkbj plus ciclosporin	Dompé farmaceutici S.p.A.	3	Completed	27 January 2022 1 October 2023
NCT05133180 PROTEGO—NGF0121 [79]	Sjögren’s dry eye	Cenegermin-bkbj	Dompé farmaceutici S.p.A.	3	Completed	14 January 2022 31 July 2023
NCT03982368 NGF0118 [80]	Dry eye	Cenegermin-bkbj	Dompé farmaceutici S.p.A.	2	Completed	1 June 2019 15 July 2020
NCT03035864 NGF0116 [81]	Refractive Surgery	Cenegermin-bkbj	Dompé farmaceutici S.p.A.	2	Completed	12 January 2017 4 September 2017
NCT03019627 [82]	Dry eye	Cenegermin-bkbj	Dompé farmaceutici S.p.A.	2	Completed	1 January 2017 31 August 2017
NCT02609165 [83]	Macular edema; retinitis pigmentosa	Cenegermin-bkbj	IRCCS San Raffaele	2	Completed	31 May 2015 28 February 2017
NCT02227147 [84]	Stage 2 and 3 neurotrophic keratitis	Cenegermin-bkbj	Dompé farmaceutici S.p.A.	2	Completed	1 February 2015 1 September 2016
NCT02101281 [85]	Dry eye	Cenegermin-bkbj	Dompé farmaceutici S.p.A.	2	Completed	20 January 2014 31 January 2015
NCT02110225 [86]	Retinitis pigmentosa	Cenegermin-bkbj	Dompé farmaceutici S.p.A.	1/2	Completed	31 January 2014 30 November 2015
NCT01756456 REPARO—NGF212 [87]	Stage 2 and 3 neurotrophic keratitis	Cenegermin-bkbj	Dompé farmaceutici S.p.A.	1/2	Completed	31 January 2013 19 May 2016
NCT02855450 [88]	Open-angle glaucoma	Cenegermin-bkbj	Dompé farmaceutici S.p.A.	1b	Completed	31 December 2016 31 May 2018
NCT05700864 [89]	Corneal pain Neuropathic pain	Cenegermin-bkbj	Tufts Medical Center	1	Terminated	1 November 2022
NCT03836859 [90]	-	Cenegermin-bkbj	Dompé farmaceutici S.p.A.	1	Completed	30 March 2018 1 June 2018
NCT01744704 [91]	-	Cenegermin-bkbj	Dompé farmaceutici S.p.A.	1	Completed	31 July 2012 31 March 2013
NCT04700657 [92]	Neurotrophic keratitis in patients with ocular GVHD	Cenegermin-bkbj	Indiana University	Not applicable	Recruiting	17 December 2020 1 December 2024
NCT04627571 [93]	Neurotrophic keratitis	Cenegermin-bkbj	University of Calfornia	Not applicable	Recruiting	23 May 2022 1 December 2024
NCT04573647 [94]	Neurotrophic ulcers	Cenegermin-bkbj	SightMD	Not applicable	Recruiting	1 October 2020 30 June 2023
NCT04552730 [95]	Neurotrophic keratitis	Cenegermin-bkbj	Stanford University	Not applicable	Completed	14 October 2020 4 May 2021

GVHD, graft versus host disease; rhNGF, recombinant human nerve growth factor.

**Table 2 biomolecules-14-00635-t002:** Comparison of clinical trials: phase 1/2 (Europe) and phase 2 (USA) [104].

	NGF0212/REPARO Phase 1 (Europe) N = 18 [99]			NGF0212/REPARO Phase 2 (Europe) N = 156 [100]			NGF0214 Phase 2 (USA) N = 48 [101]	
	rhNGF 10 μg/mL n = 7	rhNGF 20 μg/mL n = 7	Placebo n = 4	rhNGF 10 μg/mL n = 52	rhNGF 20 μg/mL n = 52	Placebo n = 52	rhNGF 20 μg/mLn = 24	Placebo n = 24
Primary NK diagnosis (%)								
Stage 2	42.9	71.4	50.0	21.0	27.0	28.0	62.5	75.0
Stage 3	57.1	28.6	50.0	31.0	25.0	24.0	37.5	25.0
Efficacy								
Complete corneal healing ratio after treatment (%), *p*-value								
4 weeks				54.9 *p* < 0.001 ^a^	58.0 *p* < 0.001 ^a^	19.6 *p* = 0.754 ^b^	54.2, *p* < 0.017 ^a^	20.8
8 weeks				74.5 *p* = 0.001 ^a^	74.0 *p* = 0.002 ^a^	43.1 *p* = 0.953 ^b^	58.3, *p* < 0.001 ^a^	12.5
15-letter gain in BCDVA change (%), *p*-value								
4 weeks				36.7 *p* = 0.097 ^a^	34.1 *p* = 0.175 ^a^	20.9 *p* = 0.008 ^b^		
8 weeks				50.0 *p* = 0.008 ^a^	41.5 *p* = 0.068 ^a^	22.5 *p* = 0.421 ^b^	13.0, *p* = 0.727 ^a^	16.7
Improvement in corneal sensitivity (%), *p*-value								
4 weeks				68.9 *p* = 0.592 ^a^	61.1 *p* = 0.835 ^a^	63.4 *p* = 0.465 ^b^		
8 weeks				78.6 *p* = 0.303 ^a^	76.3 *p* = 0.442 ^a^	68.4 *p* = 0.809 ^b^	72.2, *p* = 0.207 ^a^	60.0
Side effects (% of patients)								
Any adverse event	14.3	42.9	25.0	11.5	17.3	19.2	43.5	33.3
Eye pain	0.0	28.6	0.0	0.0	7.7	3.8	13.0	4.2
Conjunctival hyperemia	14.3	0.0	0.0	0.0	0.0	1.0	NA	NA
Eye irritation	0.0	14.3	0.0	1.9	0.0	1.9	0.0	8.3

BCDVA, best-corrected distance vision acuity; NA, not available; NGF, nerve growth factor; NK, neurotrophic keratitis; rhNGF, recombinant human NGF. ^a^ Compared with placebo. ^b^ 10 μg/mL rhNGF versus 20 μg/mL rhNGF. *p*-value ≤ 0.05 is considered statistically significant.

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
