# Peer review of "The History of Nerve Growth Factor: From Molecule to Drug"

_biomolecules, 2024, doi:10.3390/biom14060635_

Round 1

Reviewer 1 Report

Comments and Suggestions for Authors

Article. „The History of Nerve Growth Factor: From molecule to Drug is an informative and well-written overview of the history of nerve growth factor (NGF) from its discovery in the 1950s to its approval as the first topical biologic drug for the treatment of all stages of neurotrophic keratitis.

Below are some points that need attention.

The legend to Figure 1 states: "NGF binds and activates TrkA and p75NTR, resulting in … inhibition of neurite growth via MAPK". If possible, please provide an appropriate reference to this claim. Also, this is inconsistent with the picture in Figure 1. NGF has been shown to trigger activation of the MAPK signaling pathway, leading to neurite outgrowth. Please clarify

I suggest the authors delete the first paragraph in section 1.4., and just mention in the rest of the text that "Dompe is one of the leading biopharmaceutical companies in Italy". This paper clearly describes the importance of this company in the development of NGF as a drug and, in my opinion, this part has a promotional character that is superfluous in this scientific paper.

What does the abbreviation OG stand for in the discussion section, line 432? Oligodendroglioma? And is the corresponding reference cited here?

I suggest the authors mention that there is an interest in developing drugs that target NGF or its binding to receptors that may be important in autoimmune diseases, for example

minor:

I would prefer the introduction to include the non-abbreviated names of the receptors TrkA and p75NTR in addition to the abbreviations.

Line 437 in vivo and in vitro to be written in italic

Author Response

Best regards

Reviewer 2 Report

Comments and Suggestions for Authors

The manuscript by Gavioli et al. paper provides a comprehensive review of the journey of nerve growth factor (NGF), from its discovery to an approved therapeutic agent, with potential applications extending beyond ocular diseases. Overall, this is a well-written, clearly presented paper. However, to further improve the paper, providing more detailed discussions on the mechanistic insights into NGF's therapeutic (and adverse) effects would be beneficial. Modifications in the following points should further strengthen the manuscript:

The NGF signalling and effects presented in Figure 1 appear oversimplified. To fully understand the description of its therapeutic and adverse effects, additional context is necessary and should be provided both in the Figure and in the main text.

Figure 2B is not described in the main text it should be either removed or properly explained.

I suggest improving the organization and naming of review paragraphs. The inclusion of the protein production section between paragraphs discussing clinical trials disrupts the flow. Additionally, the naming conventions for paragraphs describing NGF clinical applications lack consistency (e.g., "1.3. Recombinant Human NGF ", "1.4 From Molecule to Medicinal Product" vs. "1.6. Clinical Trials of rhNGF for Neurotrophic Keratitis"). Rather than dividing the discussion by different pharmaceutical companies, the authors could consider structuring the discussion around different applications (diseases). Each disease section should conclude with a statement of whether there is currently sufficient evidence for NGF's therapeutic benefit for each disease.

L. 233-234, if available, the reasons for clinical trial suspension or discontinuation should be specified.

L. 236-251, please indicate if rhNGF for HIV-associated sensory neuropathy impacted HIV control (e.g., HIV RNA rebound or CD4+ T cell counts).

Author Response

Best regards

Reviewer 3 Report

Comments and Suggestions for Authors

In this review article, the Authors presented the sequence of events from the discovery of NGF to its journey to develop as a clinical drug! The review is well written and succeeded in delivering the intended message. 

Comments on the Quality of English Language

Quality of English language is aceeptable.

Author Response

We thank Reviewer 3 and appreciate his positive feedback on the review. The required slight English language changes have been made.